# Undernutrition and Feeding Difficulties Among Children with Disabilities in Uganda: A Cross-Sectional Study

**DOI:** 10.3390/nu18020200

**Published:** 2026-01-08

**Authors:** Zeina Makhoul, Moses Fisha Muhumuza, Bella Kyarisiima, Grace Amongin, Maria Nakibirango, Carolyn Moore, Daniella Akellot, Lutgard Musiime, Doreen Alupo, Lorna Mary Namususwa, Pamela Magero, Kate Miller, Douglas Taren

**Affiliations:** 1SPOON, Portland, OR 97210, USA; carolynm@spoonfoundation.org (C.M.); lutgard@spoonfoundation.org (L.M.); doreen@spoonfoundation.org (D.A.); lorna@spoonfoundation.org (L.M.N.); pamela@spoonfoundation.org (P.M.); katem@spoonfoundation.org (K.M.); douglas.taren@cuanschutz.edu (D.T.); 2CoRSU Rehabilitation Hospital, Kisubi P.O. Box 46, Uganda; moses.muhumuza@corsuhospital.org (M.F.M.); bella.kyarisiima@corsuhospital.org (B.K.); grace.amongin@corsuhospital.org (G.A.); maria.nakibirango@corsuhospital.org (M.N.); 3Deval Consultancy Services Ltd., Wakiso P.O. Box 256, Uganda; dakellot@gmail.com; 4Section of Nutrition, Department of Pediatrics, School of Medicine, University of Colorado, Aurora, CO 80045, USA

**Keywords:** disability, undernutrition, Uganda, cleft lip and palate, feeding difficulties, inclusion, nutrition, underweight, stunting, wasting

## Abstract

**Background/Objectives**: Inclusive nutrition services and data on children with disabilities living in low- and middle-income countries remain limited. We estimated the prevalence of undernutrition and described feeding practices and difficulties among children with disabilities ages birth to 10 years at a rehabilitation hospital in Uganda and identified barriers and opportunities for inclusive nutrition. **Methods**: This cross-sectional study enrolled 428 children. Data included demographics, weight, height, mid–upper arm circumference (MUAC), hemoglobin levels, risk for feeding difficulties, caregiver-reported feeding practices, and functional difficulties complemented by 32 caregiver and stakeholder interviews. Undernutrition was defined using WHO *z*-scores, MUAC, and anemia cutoffs. Associations were examined using Pearson’s chi-squared tests and adjusted odds ratios from logistic regression. **Results**: Over half of participants were boys (56.1%) and 65.9% were <24 months old. Common conditions included cleft lip/palate (55.4%) and cerebral palsy (38.6%). Undernutrition was prevalent: 45.2% were underweight, 38.3% stunted, 16.1% wasted (by MUAC), and 39.5% anemic. Being at risk for feeding difficulties (67.2% of children) increased the odds of underweight [AOR = 2.28 (1.23–4.24)], stunting [2.46 (1.26–4.79)], and wasting [2.43 (1.10–5.35)] after adjusting for covariates. Bottle-feeding increased the odds of stunting [3.09 (1.24–7.70)] in infants with cleft lip/palate < 12 months old. Poor access to services, food insecurity, and feeding challenges were key barriers to optimal nutrition. Most caregivers reported using practices that support responsive feeding. **Conclusions**: Reported barriers to services and high levels of undernutrition, strongly linked to feeding difficulties, underscore the need for targeted feeding interventions and better access to inclusive nutrition services in Uganda.

## 1. Introduction

Globally, children with disabilities are at high risk of undernutrition, which in turn increases their risk of delayed development, illness, and death [1]. UNICEF estimates that there are 240 million children with disabilities around the world, and those living in low- and middle-income countries are 34% more likely to be stunted and 25% more likely to be wasted compared to children without disabilities [1].

Unique nutritional risks due to factors such as increased nutrient requirements, nutrient loss, reduced food intake, and oral-motor difficulties contribute to high rates of undernutrition among this population [2,3,4]. Up to 80% of children with disabilities experience feeding difficulties [5], including difficulties in sucking, chewing, swallowing, and self-feeding, often leading to fatigue, frustration, and stress for both the child and caregiver [6].

These physiological challenges are compounded by barriers in accessing health and nutrition support, suboptimal feeding practices due to lack of knowledge or specific skills among caregivers and healthcare workers, and attitudinal, social or cultural factors [1,7]. Households raising a child with a disability are also disproportionately affected by multidimensional poverty and show higher levels of deprivation, including in nutrition [8]. In many cases, these challenges contribute to the institutionalization of children with disabilities, often as a means of accessing services and supports unavailable within their communities [1]. Yet, policies, programs, and evidence to address the specific nutritional needs of children with disabilities and feeding difficulties remain limited [9]. For example, while children with disabilities may be referenced, no specific guidance is provided on how to support the feeding and nutrition of children with disabilities with severe acute malnutrition [10], including in the updated World Health Organization (WHO) Guidelines on the Prevention and Management of Wasting and Nutritional Oedema in Infants and Children Under 5 Years [11].

Uganda has made significant efforts in disability inclusion through legislation and programs [12]. It has ratified both [13] the United Nations Convention on the Rights of Persons with Disabilities [14] and the Convention on the Rights of the Child [15], affirming the right to health care for children with disabilities. The 2023 National Disability Policy [16] also recognizes nutrition as a right for persons with disabilities. Uganda’s commitment is also reflected in programs implemented in collaboration with Organizations of Persons with Disabilities and Civil Society Organizations, focusing on capacity strengthening, socio-economic development, disability awareness, and community-based rehabilitation, among other initiatives [17].

Despite these efforts, important gaps remain. Persons with disabilities and their families in Uganda continue to face stigma, discrimination, poverty, poor access to adequate health care, support, and services [18]. The National Nutrition Plan (2020/21–2024/25) [19], while aiming to improve the nutritional status of children and other vulnerable groups, includes no specific objectives or guidance for children with disabilities. Estimates of disability prevalence among children in Uganda largely rely on the Washington Group UNICEF Child Functioning Module (CFM) Short Set that focuses on six core functional domains—seeing, hearing, walking, communication, cognition, dexterity (2–4 years old), and self-care (5–17 years old) [20]. However, the Short Set does not capture limitations in all domains and may underestimate prevalence. Surveys have reported estimates for different age ranges, making comparisons over time challenging. The 2022 Uganda Demographic and Health Survey reported rates of 5.2% among children aged 5–9 years and 2.5% among those aged 10–14 years, but did not provide estimates for children aged 2–4 years [21]. The 2024 National Population and Housing Census estimated disability prevalence at 3.3% among children aged 2–4 years and 2.1% among those aged 5–17 years, without further age disaggregation [22]. National nutrition surveys have reported high levels of undernutrition among children under five years, but have not disaggregated these data by disability [21]. Although a few recent studies from Uganda indicate high rates of undernutrition among children with specific health conditions [23,24,25,26,27,28], more data and evidence are needed to inform the inclusion of children with disabilities in nutrition policies and programs.

This study builds on existing data to (1) estimate the prevalence of undernutrition and describe feeding practices and difficulties among children with disabilities accessing care at a leading rehabilitation hospital in Uganda; (2) assess the perceived barriers to adequate nutrition and feeding among caregivers; (3) identify gaps and opportunities in feeding and nutrition services and policies for children with disabilities.

This paper frames disability using the WHO’s International Classification of Functioning, Disability, and Health (ICF), which defines disability as a “dynamic interaction between a person’s health condition, environmental factors, and personal factors [29].” Using this framework, we examine how health conditions, individual and family characteristics, and environmental barriers or enablers interact to shape nutrition and health outcomes for children with disabilities. In doing so, this paper contributes to the literature by exploring the intersection of disability, feeding difficulties, undernutrition, and contextual factors for children.

## 2. Materials and Methods

### 2.1. Study Setting and Participants

This cross-sectional study was conducted at Comprehensive Rehabilitation Services for People with Disability in Uganda (CoRSU) Rehabilitation Hospital in Kisubi, Uganda—a nonprofit hospital established in 2009 with the mission of providing surgical and rehabilitative services, including plastic, reconstructive, and orthopedic care, to people with disabilities in Uganda [30].

Children access services at CoRSU through multiple referral pathways, including government and private health facilities, non-governmental organizations, community-based rehabilitation programs, village health teams, and caregiver self-referral following outreach and awareness activities.

In addition to comprehensive cleft lip and palate care, CoRSU provides a wide range of services, including pediatric orthopedic care (such as clubfoot management), rehabilitation therapies (physiotherapy, occupational therapy, and speech therapy), provision of assistive devices, nutrition assessment and counseling, feeding support for children with feeding difficulties, and psychosocial support for caregivers.

Surgical services are provided free of charge for children under five years of age, while children aged 6–17 years receive an approximate 60% subsidy. Care is supported through donor funding and partnerships, with limited cost-sharing for selected services. When sufficient donor funding is available, all hospital services for children aged 17 years and below are offered at no cost. Service eligibility is not diagnosis-restricted, and families are not denied care due to inability to pay.

Each year, CoRSU serves more than 15,000 children with disabilities under 18 years of age from across Uganda and neighboring countries. Children receiving care at CoRSU come from diverse geographic regions and socioeconomic backgrounds, improving the geographical and socioeconomic representation of the study sample.

The study included 428 children from birth to 10 years of age. Eligible participants were children who were living in family care, had a confirmed health condition or disability as documented by a medical doctor or in the child’s medical record, and who were receiving services at CoRSU. Children were not recruited to come to the facility for the study; rather, caregivers were approached during their existing appointments and invited to participate. All children and caregivers continued to receive the standard of care based on the assessment findings conducted as part of the study. Data collection took place between July 2024 and April 2025.

In this study, the terms “children with disabilities” and “children with health conditions” refer to those with physical, intellectual, or developmental disabilities, as well as health conditions that may lead to disability. For example, although orofacial cleft lip and palate is not considered a disability in itself, if not effectively addressed, it can lead to significant and long-term difficulties, such as impaired feeding, speech, or hearing, that are classified as disabilities [31]. The terminology used in this paper aligns with the ICF framework [29], which emphasizes the interaction between health conditions and contextual factors in shaping functional outcomes and determining disability status.

### 2.2. Sample Size Estimation

A target sample size of 422 was calculated to estimate an underweight prevalence of 52.0%, with a 5% margin of error and a 95% confidence level. The estimate was based on data collected from children with disabilities participating in an ongoing nutrition and feeding program in Uganda. Underweight was selected for sample size estimation as it is a more reliable indicator of undernutrition in children with disabilities, given that weight is less error-prone and easier to measure accurately than height or length in children with physical impairments (e.g., contractures) [32,33].

### 2.3. Nutritional Status and Feeding Assessments

#### 2.3.1. Anthropometric Measurements

Measurements, including weight, length (children < 24 months old) or height (children ≥ 24 months old), and mid-upper arm circumference (MUAC) were obtained for 422 children (98.6% of the sample) using calibrated equipment, including Seca 874 electronic flat scale (Seca, Hamburg, Germany) and UNICEF wooden length/height measuring board and child MUAC measuring tape (UNICEF, New York, USA), and standardized methods [34] and based on the child’s physical ability (e.g., contractures). Length or height was not measured for 46 children who could not fully straighten their legs or stand unassisted; of these children, 84.8% had cerebral palsy, the mean age was 67.1 ± 21.4 months, and all of them had weight and MUAC measurements taken. For 30 children aged ≥ 24 months who were unable to stand but could straighten their legs, recumbent length was measured and 0.7 cm was subtracted to convert it to an equivalent standing height [35]. Measurements were recorded in *Count Me In*—a digital health app with growth, anemia, feeding, and developmental screening modules [36] used in the study to collect data.

The WHO Growth Standards and References were used to determine the level of undernutrition based on *z*-score thresholds [37]—normal (≥−2), moderate (<−2 and ≥−3), and severe (<−3)—as follows: length/height-for-age *z*-score (L/HAZ) for stunting among children ages 0–120 months; weight-for-age *z*-score (WAZ) for underweight among children ages 0–120 months; and weight-for-length/height *z*-score (WL/HZ) for wasting among children ages 0–59 months. Classification of wasting was also based on MUAC cut-offs: for ages 6–59 months: normal (≥12.5 cm), moderate (<12.5 cm and ≥11.5 cm), and severe (<11.5 cm); and for ages 60–120 months: normal (≥14.5 cm), moderate (<14.5 and ≥13.5 cm), and severe (<13.5 cm). Additionally, infants aged 6 weeks to 6 months with MUAC < 11.0 cm were classified as at risk for poor growth and development based on the updated WHO Guidelines on the Prevention and Management of Wasting and Nutritional Oedema [11].

#### 2.3.2. Anemia Screening

Hemoglobin concentration (g/dL) was measured in children aged 6 months and older using the HemoCue Hb201+ analyzer (Ängelholm, Sweden), following standard procedures [38]. A sterile lancet was used to obtain a capillary blood sample via heel prick for children under 12 months and finger prick for those 12 months and older. The third blood drop was collected in a microcuvette and analyzed immediately. Hemoglobin was not measured in children with current infections (e.g., fever, malaria) since acute-phase responses can alter hemoglobin values [39]. Both hemoglobin results and infection status were recorded in *Count Me In*.

Anemia severity was classified according to the WHO age-specific thresholds [40]: for 6–23 months: mild (9.5–10.4 g/dL), moderate (7.0–9.49 g/dL), and severe (<7.0 g/dL); for 24–59 months: mild (10.0–10.9 g/dL), moderate (7.0–9.9 g/dL), and severe (<7.0 g/dL); and for 5–11 years: mild (11.0–11.4 g/dL), moderate (8.0–10.9 g/dL), and severe (<8.0 g/dL).

#### 2.3.3. Screening for Feeding Difficulties

Screening for risk of feeding difficulties was conducted among 417 children (97.4% of the sample) following the approach previously described by Makhoul et al. [41]. Briefly, trained study team collected information in *Count Me In* from caregivers on their concerns for feeding difficulties (yes/no), assistance required for feeding (full, some, or none), tools used (breast, bottle, spoon, cup, or fingers), textures of foods offered (breast and/or formula milk, thin liquids other than milk, purees, mashed, sift and bite sized, or regular foods), duration of feeding (<10, 10–30, or >30 min), and frequent coughing or chocking (yes/no) (see Appendix A) [42,43,44,45]. When an infant was fed directly at the breast, “breast” was selected under feeding tools. Feeding of expressed breast milk was recorded by selecting “breast” along with the tool used for delivery (e.g., bottle, cup, or spoon).

Responses were compared to developmental expectations for age using built-in logic in the app to identify children at risk for delayed feeding skills and feeding difficulties. For infants 0–6 months, expected practices included full feeding assistance, feeding at the breast or by bottle, and exclusive intake of breast milk and/or infant formula. For infants 6–12 months, expected practices included full or partial assistance, use of the breast, bottle, spoon, cup, or fingers, and consumption of milk, thin liquids, purees, mashed, and soft bite-sized foods. For children 12 months and older, expected practices included partial or no assistance, feeding with the breast, spoon, cup, or fingers, and consumption of regular foods. Across all ages, feeding was expected to take 10–30 min and occur without frequent coughing or choking. Departures from these age-specific parameters were considered indicative of a feeding difficulty risk. Children with cerebral palsy were automatically categorized as at risk due to the well-documented associations of cerebral palsy with feeding difficulties [46]. The tool used is a screening instrument and therefore measures the risk of feeding difficulties, rather than diagnosing feeding difficulties themselves.

#### 2.3.4. Assessment of Feeding Practices

Adherence to recommended feeding practices was evaluated for 411 children (96.0% of the sample) using a structured questionnaire embedded in *Count Me In*. The tool captured how often caregivers (always, sometimes, or never) followed feeding practices that promote safety, efficiency, skill development, and social engagement [47,48,49,50,51] (see Appendix A). For infants < 12 months old who were breastfed, assessed practices included supporting the head during feeding (infants < 6 months old), responding to feeding cues, interacting during feeding, and, for those who were bottle-fed, additional practices assessed included maintaining nipple integrity and burping (infants < 6 months old). For children ≥ 6 months old receiving complementary feeding, practices included responsive pacing, engaging with the child, sitting at eye level, encouraging shared mealtimes, and using small, age-appropriate spoons. While these practices reflect components of responsive feeding [51], no single composite measure of responsive feeding was created, and all practices are reported individually.

### 2.4. Developmental Screening

The Washington Group UNICEF CFM full set was used to assess functional difficulties across the following domains: vision, hearing, mobility, communication/comprehension, behavior and learning (all ages); dexterity and playing (2–4 years); and self-care, remembering, focusing attention, coping with change, relationships and emotions (5–10 years) [20]. Caregivers rated each domain from “no difficulty” to “cannot do at all,” with responses of “a lot of difficulty” or “cannot do at all” indicating a functional difficulty.

### 2.5. Qualitative Data Collection

The qualitative component of the study aimed to understand families’ perceived barriers to and facilitators of nutrition and feeding for children with disabilities, as well as to identify gaps and opportunities within existing services and programs. We conducted semi-structured interviews with caregivers of children with disabilities enrolled in the study and with key stakeholders. Caregiver interviews were held in person at CoRSU, with every 10th study participant invited to participate. Interview topics included caregiver demographics; perceived benefits of nutrition and feeding; dietary diversity; feeding experiences; access to nutrition and rehabilitation services; the impact of COVID-19; sources of information and support; and recommendations for service improvement.

Key stakeholders were selected through convenience sampling and interviewed in person, by phone, or virtually. Stakeholder interviews focused on experiences working with children with disabilities, key challenges faced by families, existing services and policies, and recommendations.

Trained interviewers used a standardized guide in English or Luganda, based on participant preference. All interviews were recorded, transcribed, and translated into English as needed.

### 2.6. Statistical Analysis

#### 2.6.1. Data Management

Deidentified data from *Count Me In* were imported into STATA (StataNow/SE 19.5, StataCorp, College Station, TX, USA) for analysis. Data were cleaned and checked for completeness and outliers using exploratory data analysis methods [52].

Biologically implausible *z*-scores, defined according to WHO recommendations [35], were excluded: L/HAZ <–6 or >+6; WAZ <–6 or >+5; WL/HZ <–5 or >+5. In total, 19 children had outliers in one or more *z*-scores—L/HAZ (*n* = 13), WAZ (*n* = 7), and WL/HZ (*n* = 6).

Demographic characteristics of children with outlier *z*-scores are presented in the Appendix A. Children with L/HAZ outliers were significantly younger (14.4 ± 10.5 months) than those without (18.1 ± 22.9 months; *p* < 0.01). No significant differences were observed by age, sex, or health condition for other anthropometric indicators.

Health conditions were coded as 0 = absent or 1 = present, and the total number of conditions was determined by summing all conditions. Children were then classified into three groups based on their primary diagnosis: (1) cleft lip and/or palate, (2) cerebral palsy, and (3) other developmental disabilities (e.g., cognitive impairment, hydrocephalus, seizure disorder/epilepsy, heart disease/defect, or visual impairment). Sub-analyses were conducted for children with cerebral palsy and those with cleft lip and/or palate, as these two groups together accounted for nearly 94% of the total sample (see Appendix A).

#### 2.6.2. Quantitative Data Analysis

Descriptive analyses were conducted to summarize the characteristics of the study sample. Means and standard deviations were reported for continuous variables, and frequencies and percentages for categorical variables. Exploratory analyses were performed to assess group differences, using Pearson’s chi-squared tests for categorical variables.

Crude and adjusted logistic regression analyses were used to examine the associations between undernutrition indicators and risk for feeding difficulties (explanatory variable), with corresponding 95% confidence intervals (CIs). Variables with a *p*-value < 0.2 in the bivariate analyses were included in the multivariable models, except for age, sex, and type of health condition, which were included a priori.

Three binary logistic regression models were constructed and compared using post-estimation and goodness-of-fit tests: Model 1 estimated unadjusted odds ratios (OR); Model 2 adjusted for sex, categorical age, and type of health condition (cleft lip and/or palate/cerebral palsy/other developmental disabilities); and Model 3 additionally controlled for infection status (no/yes) and number of health conditions (one/two or more) as a proxy for overall health status. Among infants under 12 months with cleft lip and/or palate, the full model further adjusted for feeding practices, including (direct or expressed breast milk; no/yes), bottle feeding (no/yes), and adherence to best practices (less than ideal/ideal). Interaction terms between feeding risk and mode of feeding were introduced and evaluated for improvement in model fit. The events-per-variable rule of ≥10 was applied to determine the number of covariates retained in the final models to ensure model stability [53].

Variables in the models were checked for multicollinearity using Variance Inflation Factors. Model performance was assessed using the Hosmer–Lemeshow goodness-of-fit test, with a large *p*-value (>0.05) indicating good fit. The Receiver Operating Characteristic curve was used to assess the model’s discriminative ability, with an area under the curve of ≥0.70 considered acceptable. The likelihood ratio (LR) test was used to compare nested (reduced and full) models, where a *p*-value < 0.05 indicated that the full model provided a significantly better fit [54]. Outputs from post-estimation and goodness-of-fit tests are provided in the Appendix A.

Statistical significance was set at a *p*-value of <0.05 for all analyses.

#### 2.6.3. Qualitative Data Analysis

Qualitative analysis methods were adapted from the process described by Coates et al. [55], with coding and identification of themes informed by Naeem et al. [56]. Researchers divided the transcripts and each conducted a preliminary coding of 5 transcripts. They then met and agreed on a preliminary list of codes, which were defined in the codebook. Researchers then coded all transcripts, applying the agreed-upon codes and any new codes that emerged. Data were then entered into a spreadsheet organized by respondent and by code. Once all codes were entered, each researcher reviewed all data and noted any questions or areas of disagreement. The two researchers met and resolved all questions. Then, researchers worked together to combine codes and generate themes and findings. Themes were also validated with the full study team.

Family interviews, linked with children enrolled in the study, were scored on three domains—access to services, access to food, and feeding difficulties—using a scale of 0 = insufficient access or current challenges or 1 = sufficient access or no current challenges. Scores across the three domains were summed to generate an overall challenge score ranging from 0 (challenges in all domains) to 3 (sufficient support in all domains). Pearson’s chi-squared tests were used to compare undernutrition indicators across the different score categories (Appendix A).

### 2.7. Ethical Considerations

The study received ethical approval from the Mildmay Uganda Research Ethics Committee, Kampala, Uganda (reference # MUREC-2022-147), the Uganda National Council for Science and Technology Research Ethics Committee, Kampala, Uganda (reference # HS2826ES), and the Institutional Review Board at St. Catherine’s University, St. Paul, Minnesota (reference # 1900). The Uganda Ministry of Gender, Labor, and Social Development, responsible for overseeing the well-being of children with disabilities, also granted permission to collect data. Informed consent and assent were obtained from all adult participants and children over the age of 8 years, respectively.

The study was conducted in accordance with the Declaration of Helsinki [57].

## 3. Results

### 3.1. Characteristics of Study Participants

A total of 428 children were included in the study, with a mean age (±SD) of 23.1 ± 27.7 months at the time of anthropometric assessment (Table 1). Most children (69.4%) were younger than 24 months, and 43.9% were female. The primary health conditions were cleft lip and/or palate (55.4%), cerebral palsy (38.9%), and other developmental disabilities (6.1%). Most participants presented with a single condition; however, 13.6% had two or more conditions, typically a combination of another condition alongside cleft lip and/or palate or cerebral palsy. Among children aged 2–4 years, 76.8% experienced functional difficulties in at least one domain, increasing to 85.4% among those aged 5–10 years. The most frequently reported domains of difficulty were walking, communication, fine motor or self-care, and learning.

### 3.2. Nutritional and Feeding Status

Among children < 10 years old, 45.2% were underweight, 38.3% were stunted, 16.1% were wasted based on MUAC cutoffs, and 39.5% were anemic (Table 2). Among children < 5 years old, one in four (25.6%) were wasted based on WL/HZ. According to the updated WHO Guidelines on the Prevention and Management of Wasting and Nutritional Oedema, which classify infants aged 6 weeks to 6 months as “at risk for poor growth and development” [11], 5% of infants fell into this category.

Two-thirds of children (67.2%) were identified as at risk for feeding difficulties through the study’s screening tool. Those at risk had a higher prevalence of underweight (50.6% vs. 35.8%, *p* = 0.005), stunting (43.7% vs. 31.0%, *p* = 0.02), and wasting based on WL/HZ (32.0% vs. 15.1%, *p* = 0.001) or MUAC cutoffs (18.4% vs. 2.1%, *p* = 0.023) compared to those without risk of feeding difficulties. Similar patterns were observed when feeding difficulties were reported by caregivers, particularly for underweight and wasting based on WL/HZ (Table 2).

Children with multiple health conditions had a higher prevalence of stunting than those with a single condition (59.5% vs. 35.4%, *p* = 0.003). Reported infection was associated with a higher prevalence of underweight (52.5% vs. 40.6%, *p* = 0.022) and wasting based on MUAC (23.9% vs. 10.8%, *p* = 0.004), but not WL/HZ. Wasting among children under 5 years based on WL/HZ was significantly higher among those with other developmental disabilities compared to children with cleft lip and/or palate or cerebral palsy; however, this finding is based on a small sample (*n* = 12) in the other developmental disabilities group. No significant differences in undernutrition indicators were observed by age, sex, or reported coughing or choking (Table 2).

While underweight and wasting (based on WL/HZ or MUAC) were higher among children with functional difficulties in at least one domain, these differences were not statistically significant (Table 2). However, children with cerebral palsy who had mobility difficulties showed significantly higher levels of stunting (50.0% vs. 14.3%, *p* = 0.020) and MUAC-defined wasting (22.2% vs. 0%, *p* = 0.032), with underweight (62.2% vs. 38.9%, *p* = 0.073) and anemia (35.7% vs. 7.7%, *p* = 0.052) showing borderline significance (Appendix A).

### 3.3. Reported Feeding Practices

Ten feeding practices were assessed—five for infants under 12 months and five for children aged 6 months and older—with most caregivers reporting that recommended practices were always followed (Figure 1).

Among infants < 6 months, nearly all caregivers (96.8%) reported always supporting the infant’s head during feeding and most caregivers of infants < 12 months reported always following feeding cues (84.7%). However, only about half (56.5%) reported always interacting with their infants while feeding. Among bottle-fed infants < 12 months, 27.2% of caregivers did not consistently keep bottle nipples intact, and among infants < 6 months, 30.3% were not routinely burped (Figure 1a).

For children 6–23 months old, most caregivers (94.2%) reported always following their child’s feeding cues with appropriate pacing, and 89.4% consistently used small, age-appropriate spoons. Two-thirds of caregivers always sat at eye level (69.6%) and interacted with their child (65.3%) during meals. The least commonly reported practice was having the child eat with others—only half (51.4%) of caregivers reported that their child always joined others at mealtime (Figure 1b).

No significant differences in adherence to recommended feeding practices were found by the child’s sex or health condition. However, adherence varied significantly by age: on average, caregivers who had younger children (6.9 ± 6.5 months) were more likely to follow recommended feeding practices compared to those who had older children (8.7 ± 6.2 months, *p* < 0.05). Infants < 6 months had the highest proportion of caregivers adhering to ideal practices (43.1%) compared to older age groups (6–11 years: 22.5%; 12–23 months: 34.7%; *p* < 0.05).

### 3.4. Association of Risk for Feeding Difficulties with Undernutrition

Children identified as at risk for feeding difficulties had significantly higher odds of underweight (AOR = 2.28, 95% CI: 1.23, 4.24, *p* = 0.009), stunting (AOR = 2.46, 95% CI: 1.26, 4.79, *p* = 0.008), and wasting based on WL/HZ (AOR = 2.43, 95% CI: 1.10, 5.35, *p* = 0.028) compared to those not at risk, after adjusting for age, sex, type of condition, number of conditions, and reported infection (Model 3, Table 3). The increasing strength of associations across Models 1–3 for stunting and between Models 1 and 2 for underweight—evidenced by the LR test and the larger ORs—suggests that covariates may have attenuated the unadjusted relationships, indicating that risk for feeding difficulties is an independent predictor of undernutrition.

Overall, children with reported infection (e.g., malaria, fever) at the time of assessment had greater odds of being underweight (AOR = 1.53, 95% CI: 0.99–2.35, *p* = 0.053) (Appendix A), and those with multiple health conditions were more than three times more likely to be stunted compared to children with a single condition (AOR = 3.30, 95% CI: 1.62–6.73, *p* = 0.001) (Appendix A), after adjusting for other covariates in the model.

Among infants < 12 months old with cleft lip and/or palate, risk for feeding difficulties was associated with approximately threefold higher odds of underweight (AOR = 2.70, 95% CI: 1.13–6.48, *p* = 0.026) and stunting (AOR = 3.27, 95% CI: 1.30–8.21, *p* = 0.012), after controlling for age, sex, breastfeeding (direct or expressed breast milk), bottle feeding, feeding practices, and number of conditions (Figure 2). The strength of these associations increased with adjustment for covariates based on the LR test and an increase in the magnitude of the OR (Appendix A), highlighting risk for feeding difficulties as an independent predictor of undernutrition.

Breastfeeding, whether direct or expressed breast milk, showed a protective odds ratio against underweight, although this was only borderline significant (AOR = 0.46, 95% CI: 0.21–1.03, *p* = 0.059), while bottle feeding significantly increased the odds of stunting (AOR = 3.09, 95% CI: 1.24–7.70, *p* = 0.015) (Figure 2). Adherence to recommended infant feeding practices appeared associated with undernutrition but did not reach statistical significance (Figure 2). Infants with cleft lip and/or palate who also had additional health conditions were nearly five times more likely to be stunted than those with cleft lip and/or palate alone (AOR = 5.10, 95% CI: 1.42–18.34, *p* = 0.013), though the wide confidence interval suggests considerable uncertainty around the strength of this association (Appendix A).

Inclusion of interaction terms between feeding risk and mode of feeding did not alter the associations between feeding risk and undernutrition.

### 3.5. Qualitative Results

Qualitative interviews were conducted with 16 parents (13 mothers, 3 fathers) of children with disabilities, most of whom (*n* = 14) had cleft lip and/or palate, and with 16 key stakeholders, including representatives from government or multilateral organizations (*n* = 2), residential care facilities (*n* = 9), and civil society, nonprofit, and faith-based organizations (*n* = 5). Families reported barriers and enabling factors across three domains: access to services, feeding, and access to food. Among families, 5 reported sufficient access to services and food or no current challenges, 6 experienced a mix of barriers and enabling factors, and 5 faced challenges across all domains. There was a trend toward lower undernutrition (underweight and WL/HZ-based wasting) and reduced risk for feeding difficulties with more sufficient access, though these associations were not statistically significant, likely due to small sample size (Appendix A).

Themes from families and stakeholders were largely consistent, highlighting similar barriers, enabling factors, and recommendations (Table 4).

#### 3.5.1. Barriers

Families of children with disabilities reported significant challenges to their child’s nutrition and feeding, including a lack of access to food, poor access to services, feeding difficulties, stress, and stigma (Table 4). Families facing challenges in access to food mentioned cost and availability as main barriers, as well as challenges accessing a sufficient variety of food to provide a diverse diet or offer textures suitable for children with feeding difficulties. Key stakeholders described poverty as a major cause of food insecurity, and said that conflict and instability contribute to food insecurity in some areas.

Barriers in access to services were primarily related to cost and distance to travel; key stakeholders also noted a lack of specialized care. Families described a range of feeding difficulties, including food coming out of a child’s nose when feeding or a need for surgery. Many families interviewed were preparing for their child to receive a surgical cleft repair.

Families also reported feeling stress around having a child with a disability, and experiencing stigma and exclusion in their communities. Stakeholders described high levels of stigma and lower family support for families of children with disabilities, which could exacerbate stress and lead to children being placed in residential care institutions. Over half of the families reported having experienced impacts from the COVID-19 pandemic, including food insecurity, economic hardship, and a lack of transportation.

#### 3.5.2. Enabling Factors

Enabling factors included sufficient access to food and access to services, knowledge of nutrition and feeding practices, children feeding well, and economic stability. Families with sufficient access to services noted access to transportation, convenient location of services, and the ability to afford care as supportive factors. Families who had received nutrition counseling reported a high level of confidence in their ability to improve their child’s nutrition and prevent future malnutrition. All families reported good knowledge of the importance of nutrition and safe feeding. Children having good appetites and feeding well, as well as feeling bonded to and able to respond to their children, were mentioned by parents as factors that made feeding easier. Ability to buy food, support from spouses to purchase food, and breastfeeding were all named as factors that supported families to have access to food.

Stakeholders also said that home gardens were an important factor in food security. They also described some existing government programs and a policy framework that should be expanded to support families in improving the nutritional status of children with disabilities.

#### 3.5.3. Recommendations

Families recommended improving access to services, including healthcare; raising awareness of disability and disability-inclusive nutrition practices; support for families, including income generation opportunities and psychosocial support. They also emphasized the need for free or low-cost services and for nutrition services for children with disabilities to be available at more health facilities.

Key stakeholders also recommended increasing access to research and data on disability and nutrition; improving community outreach; and improving health services, including expanding services, reducing costs, and improving access to assistive technology for children with disabilities.

## 4. Discussion

This study contributes to the evidence on undernutrition among children with disabilities in Uganda by using a mixed-methods approach to integrate findings on nutritional outcomes, functional and feeding difficulties, feeding practices, and caregiver and stakeholder perspectives on support, gaps, and opportunities. The study’s key findings show that more than two-thirds (68.6%) of children with disabilities receiving services at a leading rehabilitation hospital in Uganda were moderately or severely undernourished, as indicated by stunting, underweight, or wasting, with those at risk for feeding difficulties (67.2%) being particularly vulnerable. Among infants with cleft lip and/or palate, the results emphasize the critical importance of early interventions, including breastfeeding support and nutrition education on safe alternative feeding options (e.g., breast milk expression, the use of cup or spoon) when breastfeeding is not possible, to improve nutrition outcomes.

The study further highlighted the complex interplay between health conditions and contextual factors, consistent with the ICF framework: food insecurity, limited access to services, caregiver stress, stigma, and feeding challenges emerged as barriers, while food availability, access to services, economic stability, and caregiver knowledge and skills served as enabling factors. Together, these findings underscore the urgent need for inclusive, community-based nutrition and feeding programs and policies that address both biological and environmental determinants of undernutrition among children with disabilities in Uganda and similar settings.

While this study does not evaluate interventions, existing frameworks offer relevant direction, including UNICEF’s twin-track approach [58], which emphasizes mainstreaming disability inclusion across basic services, while also strengthening disability-specific supports. Similarly, recommendations by Klein et al. [9] to strengthen health and nutrition systems, provide direct support to families, strengthen advocacy, and generate evidence on effective interventions that align closely with our findings, highlighting the need for coordinated action at both facility and systems levels.

### 4.1. Comparison with National Averages

Stunting, underweight, and wasting among the overall population of children under five in Uganda have declined nationally from 28.9%, 10.5%, and 3.6% in 2016 to 24.4%, 9.7%, and 3.2% in 2022, respectively [21]. These improvements have been largely attributed to improved infant and young child feeding practices, scaled-up nutrition-specific interventions, better access to water, sanitation, and hygiene (WASH) services, strengthened nutrition information systems, and overall socioeconomic development [59,60]. In contrast, the prevalence of stunting (38.3%), underweight (44.0%), and wasting (25.6%) among children under five in our sample was substantially higher—approximately 1.6, 4, and 8 times the national averages, respectively—demonstrating persistent inequities in nutrition outcomes among children with disabilities.

Conversely, the prevalence of anemia among children aged 6–59 months (41.5%) in our study was lower than the most recent WHO estimate of 51.7% [61]. This difference may be partly due to not adjusting hemoglobin concentrations for altitude, as recommended by the WHO [40], potentially underestimating anemia prevalence [62]. Additionally, as participants were under clinical care at CoRSU, some may have benefited from prior or ongoing nutritional support—such as nutrition education, food packages or iron and micronutrient supplementation—potentially lowering anemia rates. Although no significant differences in anemia by age were observed, a decreasing trend with increasing age was noted, consistent with broader evidence [63].

### 4.2. Comparison with Evidence from Uganda and the Region

Our results align with evidence from low- and middle-income countries showing that children with disabilities face a substantially higher risk of undernutrition. A meta-analysis of 17 studies from these settings demonstrated that children with disabilities were three times more likely to be underweight and twice as likely to be stunted or wasted compared to peers without disabilities [4]. More recent cross-country analyses using UNICEF’s Multiple Indicator Cluster Survey (MICS) data from 30 low- and middle-income countries similarly revealed significant disparities in stunting, wasting, and underweight among children with disabilities aged 2–4 years [64]. Similarly, a recent analysis of MICS data from 15 sub-Saharan African countries found that high rates of functional disability and developmental delay were associated with elevated stunting and underweight in children 36–59 months old, underscoring the interrelated nature of developmental, functional, and nutritional vulnerabilities in young children [65]. Although these analyses included several countries from Eastern and Southern Africa, data from Uganda were notably absent, highlighting the need for country-level studies to better understand local patterns and determinants, inform national policies and resource allocation, and enable tracking of progress.

Consistent with these global findings, several recent studies from Uganda and the region report similarly high levels of undernutrition among children with disabilities, particularly those with cleft lip and/or palate and cerebral palsy—the two main conditions represented in our study. Comparable regional data on risk of feeding difficulties are limited, but the prevalence in our sample (67.2%) aligns with existing estimates for children with disabilities [9].

#### 4.2.1. Children with Cleft Lip and/or Palate

Hospital- and program-based studies of children with cleft lip and/or palate in Uganda support our findings regarding feeding challenges and poor nutritional status. Two studies at CoRSU Rehabilitation Hospital, conducted five years apart (2013–2014 and 2018–2019), reported higher levels of undernutrition among infants [28] and children under 5 years old [25] with a cleft lip and/or palate compared to our study. Tungotyo et al. found that approximately 57% of infants had moderate to severe wasting (WH/LZ) [28], while Mbuga et al. reported very high rates of stunting (44.4%), underweight (57.4%), and wasting (53%) before surgical repair, with significant improvements postoperatively [25]. In contrast, the wasting levels in our study were substantially lower—20.1% among infants under 12 months and 19.3% among children under 5, roughly one-third of the levels reported in previous studies. The lower prevalence of wasting in our sample may reflect ongoing improvements in nutrition services at CoRSU over the years, including the provision of take-home nutrition packages to address food insecurity and capacity strengthening for facility staff. Differences in the timing of surgical interventions, prior nutritional support, or the inclusion of children with less severe clefts may also have contributed, although we did not have data on cleft type or prior surgical repair or nutrition interventions to confirm this.

Infants with cleft lip and/or palate are highly vulnerable to undernutrition because the cleft impairs suction, making breastfeeding or bottle-feeding difficult and often insufficient to meet nutritional needs [66]. Without specialized feeding support and counseling, these challenges can lead to growth faltering and poor nutritional outcomes [67]. If infants with cleft lip and/or palate are unable to breastfeed effectively, they may require alternative feeding methods such as cup, spoon, or tube feeding. In settings where good hygiene can be maintained, specialized bottles or infant feeders may also be used [68].

At CoRSU, caregivers are routinely counseled to hand-express breast milk when direct breastfeeding is difficult, feeding infants via cup, spoon, or specialized bottle when available. Formula feeding is considered a secondary option, particularly for families with food insecurity and limited access to safe water. Exclusive expressed milk feeding typically continues until the age of 3 months, after which caregivers transition to mixed feeding or formula. In our sample of infants younger than 3 months (*n* = 84), 89.3% received breast milk and/or infant formula exclusively (with no other liquids). Among these infants, 36.0% were fed directly at the breast, 26.7% received expressed breast milk via bottle (most commonly) or cup, and 37.3% were fed infant formula using bottle (most commonly), cup, or, in one case, spoon. With increasing age, the proportion of infants receiving formula increased, while the proportion receiving breast milk declined.

In our sample, only 43.9% of infants under 6 months old were exclusively breastfed (either directly or using expressed breast milk)—far below the national rate of 94% [21]—and just 16.1% of those aged 6–11 months continued breastfeeding, likely reflecting feeding difficulties. Caregivers who reported feeding challenges or whose infant was at risk for such difficulties were less likely to breastfeed. Similar findings were reported by Nabatanzi et al. [69], who linked ineffective feeding techniques and early bottle use to growth faltering among infants with oral clefts receiving care at CoRSU. In our study, breastfeeding (direct or expressed breast milk) appeared protective (though nonsignificant after adjusting for feeding difficulties), whereas bottle-feeding—reported by 60.2% of caregivers—was associated with higher odds of undernutrition among infants under 12 months. Notably, one in four caregivers who bottle-fed their infants reported altering bottle nipples, an unsafe and unhygienic practice. The type of bottle used (regular or specialized) was not reported.

Together, these findings highlight the need for early feeding interventions, promotion and support of breastfeeding, and ongoing caregiver counseling on breast milk expression, alternative feeding methods (e.g., cup, spoon), and safe bottle use (e.g., nipple care and hygiene) when bottle-feeding is unavoidable.

#### 4.2.2. Children with Cerebral Palsy

In rural eastern Uganda, approximately two-thirds (64%) of children and adolescents with cerebral palsy were malnourished (stunted, wasted, or underweight) [26], comparable to the level of undernutrition (69.8%) we found in our sample. However, the proportions of underweight (52.5% vs. 70.7%), stunting (40.9% vs. 74.0%), and wasting based on MUAC (19.3% vs. 22.4%) in our study were lower than those previously reported among children with cerebral palsy aged 0–10 years in Zambia [41]. This lower prevalence may partly reflect the younger age distribution of participants in our sample (42% under 24 months versus 18% in the Zambia cohort), as undernutrition tends to worsen with age among children with cerebral palsy [70]. Similarly high rates of stunting (71.4%) and underweight (69.2%) have been reported among children with cerebral palsy in residential care institutions in Zambia [71], as well as in multi-country data from Bangladesh, Indonesia, Nepal, and Ghana, where 72–98% of children with cerebral palsy were undernourished [72].

Across studies, feeding difficulties consistently emerged as a key risk factor for undernutrition. Feeding difficulties and inability to self-feed were strong predictors of undernutrition among children with cerebral palsy in rural eastern Uganda [26], while in Lusaka, Zambia, feeding challenges and caregiver practices were major contributors to poor nutrition [73]. Although these studies focused on cerebral palsy, our findings show that feeding difficulties overall more than doubled the odds of underweight, stunting, and wasting among children with disabilities.

Greater motor impairment severity among children with cerebral palsy—measured using the Gross Motor Function Classification System (GMFCS)—has been a strong predictor of undernutrition [26,73,74]. The CFM used in this study similarly assesses functional mobility limitations. Consistent with findings from GMFCS-based studies, children with cerebral palsy in our sample who had mobility difficulties were more likely to be undernourished, underscoring that functional mobility limitations—whether assessed clinically or through caregiver report—are key predictors of poor nutritional status.

We found high levels of severe undernutrition among children with cerebral palsy (31.7% underweight, 23.6% stunted, and 6.0% wasted), and 13% of children had both stunting and wasting. Evidence from other studies indicates that such severe and coexisting forms of undernutrition substantially increase mortality risk among children with cerebral palsy [70,75], further underscores the need to focus on this population as a group at high risk of undernutrition and its adverse consequences.

### 4.3. Interplay Between Health Conditions and Contextual Factors

Recent studies in Uganda underscore that national nutrition gains have not been equitably realized for children with disabilities. Households with children with disabilities had lower access to nutrition, health, and WASH services and higher rates of malnutrition than households without children with disabilities [27]. Likewise, children with cerebral palsy in Uganda faced insufficient access to rehabilitation, education, and assistive devices [23]—factors that limit mobility and self-care, the two functional domains most affected in our sample. In our study, families reporting better access to food, services, and nutrition and feeding support tended to have better nutritional outcomes, although our sample size limited quantitative validation. While feeding practices could still be improved, they were generally positive, indicating that caregiver training alone is unlikely to resolve the high rates of undernutrition observed in this population. Together, these studies support our conclusion that the persistently high burden of undernutrition in this population stems from biological vulnerabilities and systemic inequities in care and service access. Mitigating the effects of undernutrition among children with disabilities, particularly children with feeding difficulties, requires early identification of feeding difficulties, caregiver-centered counseling on safe and responsive feeding using locally available nutrient-dense foods, and integration of nutrition support within routine rehabilitation and primary healthcare services [9].

### 4.4. Study Limitations

This study has several limitations. First, because participants were recruited at CoRSU—a specialized rehabilitation hospital that primarily serves specific disability groups—selection bias is possible, and findings may not represent all children with disabilities in Uganda. Second, qualitative interview subjects were not necessarily reflective of the entire study sample, with most families having a child with a cleft lip and/or palate and most key stakeholders representing civil society. Third, measurement errors may also have occurred, particularly for children with contractures, although trained nutritionists with relevant experience conducted all assessments and they were prompted to answer questions about the child’s physical abilities. Fourth, hemoglobin concentrations were not adjusted for altitude as recommended by the WHO [40], which may have led to an underestimation of anemia [62]. Fifth, additional details on the child’s health condition—such as cleft type and surgical repair status for those with cleft lip and/or palate, and the level of motor impairment for children with cerebral palsy using GMFCS [76]—would have strengthened the interpretation of the findings and facilitated comparison with existing literature. Sixth, although the tools used to assess feeding best practices and screen for feeding difficulties were informed by evidence and clinical expertise, they have not undergone formal validation. Seventh, the small sample size within certain subgroups (e.g., type of health condition) may have contributed to wide confidence intervals and reduced the precision and stability of the estimated associations. Finally, the study did not capture certain potential determinants of undernutrition, including caregiver education, socioeconomic status, and duration of engagement with CoRSU services.

### 4.5. Implications for Practice and Research

Taken together, our findings, alongside regional and global evidence, suggest three clear implications for practice and research:Mainstream nutrition programs must be adapted to include children with disabilities, and targeted services must be available, incorporating feeding assessments, early feeding support, caregiver counseling, and linkages to rehabilitation services. In Uganda, such integration could be operationalized through existing Ministry of Health nutrition services, community-based rehabilitation structures, and referral linkages between district health facilities and specialized rehabilitation centers, ensuring disability-inclusive nutrition support without creating parallel systems.National data should be disaggregated by disability, used in health planning and to inform nutrition policies and strategies, ensuring nutrition interventions reach those most at risk for undernutrition.Future research should examine potential determinants of undernutrition (e.g., maternal education, food insecurity, socioeconomic status), quantify the relative contribution of feeding difficulties and contextual factors (i.e., barriers and enablers) to undernutrition in children with disabilities (for example, via longitudinal cohorts or intervention trials), and investigate the effective measures of reducing barriers to nutrition services, so that interventions can be prioritized and evaluated.


## 5. Conclusions

Children with disabilities in Uganda, as well as other low- and middle-income countries, remain disproportionately affected by undernutrition. Coordinated action by governments, civil society, organizations of persons with disabilities, and international agencies that addresses the multiple barriers experienced by children with disabilities and their families will be essential to guarantee their right to proper nutrition, health, and well-being.

## Figures and Tables

**Figure 1 nutrients-18-00200-f001:**
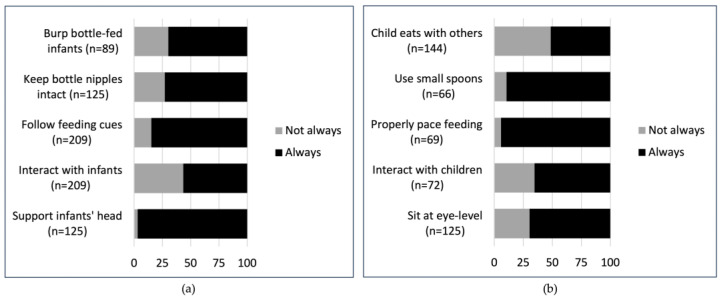
Feeding practices among children with disabilities 0–23 months old enrolled in the study: (**a**) How often caregivers followed best practices for infants 0–11 months old. Burping and supporting the infant’s head are reported for infants < 6 months old; (**b**) How often caregivers followed best practices for children 6–23 months old.

**Figure 2 nutrients-18-00200-f002:**
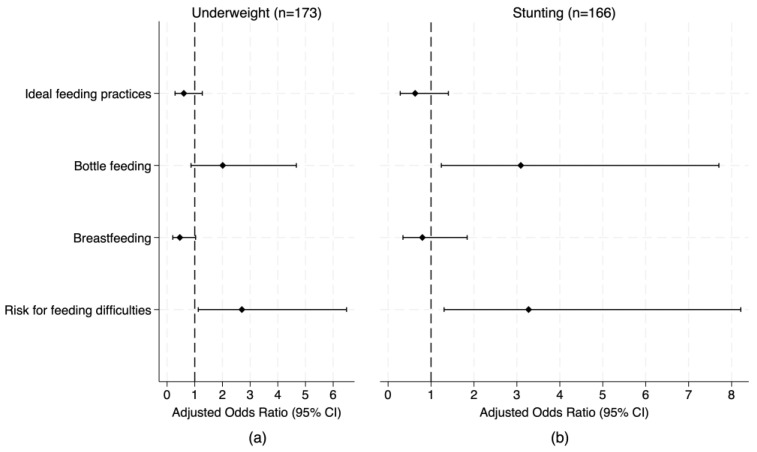
Association of risk for feeding difficulties and feeding practices with undernutrition in infants with cleft lip/palate younger than 12 months old: (**a**) Underweight; (**b**) Stunting. Logistic regression was used to identify associations of risk for feeding difficulties and feeding practices with undernutrition indicators. Models adjusted for sex, age (<6 months/6–11 months), and number of conditions (one/two or more).

**Table 1 nutrients-18-00200-t001:** Characteristics of children with disabilities enrolled in the study (*n* = 428).

Characteristic	*N*	*n* (%)
Sex	428	
Female		188 (43.9)
Male		240 (56.1)
Age at anthropometric assessment	422	
Mean age (months)		23.1 ± 27.7
Age range (months)		0.03–119.8
Age by group		
<6 months		137 (32.5)
6–11 months		83 (19.7)
12–23 months		73 (17.3)
24–59 months		81 (19.2)
60–120 months		48 (11.4)
Age at hemoglobin testing (≥6 months old)	274	
Mean age (months)		32.8 ± 28.5
Age range (months)		6.0–117.8
Age by group		
6–11 months		48 (27.9)
12–23 months		44 (25.6)
24–59 months		50 (29.1)
60–120 months		30 (17.4)
Health conditions ^1^	428	
Cleft lip and palate		237 (55.4)
Cerebral palsy		166 (38.9)
Developmental disabilities other than cerebral palsy		
Cognitive impairment		14 (3.3)
Seizure disorder/epilepsy		46 (10.8)
Hydrocephalus		12 (2.8)
Heart disease/defect		2 (0.5)
Visual impairment		1 (0.2)
Other health conditions		
Low birth weight (<2.5 kg)		4 (0.9)
Premature birth		2 (0.5)
Other ^2^		9 (2.1)
Health condition groups ^3^	428	
Cleft lip and palate		237 (55.4)
Cerebral palsy		165 (38.6)
Developmental disabilities other than cerebral palsy		26 (6.1)
Number of health conditions per child	428	
One		370 (86.4)
Two or more		58 (13.6)
Functional difficulties in children 2–4 years old	82	
Difficulty in at least one domain Breakdown by domain		63 (76.8)
Mobility (walking)		53 (64.6)
Communication		38 (46.3)
Fine motor		33 (40.2)
Learning		23 (28.1)
Playing		21 (25.6)
Controlling behavior		7 (8.5)
Seeing		6 (7.3)
Hearing		3 (3.7)
Functional difficulties in children 5–10 years old	48	
Difficulty in at least one domain		41 (85.4)
Breakdown by domain		
Mobility (walking)		32 (66.7)
Self-care (e.g., feeding, dressing)		25 (52.1)
Communication		23 (47.9)
Learning		19 (39.6)
Remembering		17 (35.4)
Making friends		15 (31.3)
Concentrating		14 (29.2)
Accepting change		14 (29.2)
Controlling behavior		13 (27.1)
Anxiety		4 (8.3)
Seeing		3 (6.3)
Hearing		3 (6.3)
Depression		2 (4.2)

^1^ Health conditions were confirmed by a medical doctor or the child’s medical record during the study. ^2^ Conditions under “Other” were not specified. ^3^ Children with cerebral palsy or cleft lip/palate were placed in their respective group regardless of the presence of other conditions. One child with cerebral palsy and cleft lip/palate (age = 24.6 months) was placed in the cerebral palsy group. Children with cleft lip/palate who have a developmental disability other than cerebral palsy were placed in the cleft lip/palate group.

**Table 2 nutrients-18-00200-t002:** Prevalence of undernutrition among children with disabilities from birth to 10 years enrolled in the study.

	Underweight ^1^		Stunting ^1^		Wasting ^1,2^				Anemia ^3^	
	WAZ (0–120 Months) *N* = 414		L/HAZ (0–120 Months) *N* = 347		WL/HZ (0–59 Months) *N* = 324		MUAC in cm (6–120 Months) *N* = 279		Hemoglobin (g/dL) (6–120 Months) *N* = 172	
Variables	*n* (%)	*p*-Value	*n* (%)	*p*-Value	*n* (%)	*p*-Value	*n* (%)	*p*-Value	*n* (%)	*p*-Value
Undernutrition		--		--		--		--		--
No	227 (54.8)		214 (61.7)		241 (74.4)		234 (83.9)		104 (60.5)	
Mild/Moderate	78 (18.8)		62 (17.9)		46 (14.2)		30 (10.8)		68 (39.5)	
Severe	109 (26.3)		71 (20.5)		37 (11.4)		15 (5.4)		0 (0)	
Sex		0.152		0.643		**0.050**		0.173		0.185
Female	75/182 (41.2)		60/162 (37.0)		30/147 (20.4)		16/125 (12.8)		27/79 (34.2)	
Male	112/232 (48.3)		73/185 (39.5)		53/177 (29.9)		29/154 (18.8)		41/93 (44.1)	
Age		0.247		0.233		0.119		0.817		0.455
<6 months	60/135 (44.4)		47/130 (36.2)		28/131 (21.4)		--		--	
6–11 months	29/81 (35.8)		26/78 (33.3)		17/80 (21.3)		12/82 (14.6)		23/48 (47.9)	
12–23 months	32/71 (45.1)		30/58 (51.7)		20/59 (33.9)		10/72 (13.9)		17/44 (38.6)	
24–59 months	40/79 (50.6)		20/55 (36.4)		18/54 (33.3)		15/79 (19.0)		19/50 (38.0)	
60–120 months	26/48 (54.2)		10/26 (38.5)		--		8/46 (17.4)		9/30 (30.0)	
Health conditions		0.061		0.183		**0.001**		0.074		0.112
Cleft lip/palate	94/231 (40.7)		85/220 (38.6)		42/218 (19.3)		10/103 (9.7)		33/68 (48.5)	
Cerebral palsy	83/158 (52.5)		45/110 (40.9)		36/94 (38.3)		29/150 (19.3)		29/90 (32.2)	
Other developmental disabilities	10/25 (40.0)		3/17 (17.7)		5/12 (41.7)		6/26 (23.1)		6/14 (42.9)	
Number of health conditions		0.073		**0.003**		0.771		0.167		0.976
One	156/359 (43.5)		108/305 (35.4)		72/284 (25.4)		35/236 (14.8)		57/144 (39.6)	
Two or more	31/55 (56.4)		25/42 (59.5)		11/40 (27.5)		10/43 (23.3)		11/28 (39.3)	
Reported feeding difficulties ^4^		**0.019**		0.174		**0.002**		0.056		0.219
No	79/210 (37.6)		71/199 (35.7)		32/197 (16.2)		9/97 (9.3)		27/63 (42.9)	
Yes	22/38 (57.9)		15/31 (48.4)		12/30 (40.0)		6/26 (23.1)		8/13 (61.5)	
At risk for feeding difficulties ^5^		**0.005**		**0.020**		**0.001**		**0.023**		0.718
No	48/134 (35.8)		39/126 (31.0)		19/126 (15.1)		1/34 (2.9)		8/23 (34.8)	
Yes	138/273 (50.6)		94/215 (43.7)		63/197 (32.0)		44/239 (18.4)		55/142 (38.7)	
Reported coughing and choking		0.058		0.801		0.189		0.239		0.570
No	135/313 (43.1)		107/272 (39.3)		60/253 (23.7)		28/190 (14.7)		47/127 (37.0)	
Yes	51/94 (54.3)		26/69 (37.7)		22/70 (31.4)		17/83 (20.5)		16/38 (42.1)	
Reported infection		**0.022**		0.744		0.249		**0.004**		--
No	108/266 (40.6)		88/234 (37.6)		51/218 (23.4)		18/166 (10.8)		-- ^6^	
Yes	75/143 (52.5)		43/109 (39.5)		30/102 (29.4)		26/109 (23.9)		--	
Functional difficulties		0.090		0.911		0.129		0.108		0.988
No	10/27 (37.0)		8/22 (36.4)		3/16 (18.8)		2/27 (7.4)		6/17 (35.3)	
Yes	57/103 (55.3)		23/61 (37.7)		16/40 (40.0)		21/101 (20.8)		22/62 (35.5)	

L/HAZ: Length/Height-for-age *z*-score; MUAC: Mid-upper arm circumference; WAZ: Weight-for-age *z*-score; WL/HZ: Weight-for-length/height *z*-score. Statistical analyses using Pearson’s chi-squared. *p*-values shown in bold are statistically significant (<0.05). ^1^ Classification of underweight, stunting, and wasting (WL/HZ) was based on *z*-score thresholds: normal (≥−2), moderate (<−2 and ≥−3), and severe (<−3). ^2^ Classification of wasting was also based on MUAC cut-offs: for ages 6–59 months: normal (≥12.5 cm), moderate (<12.5 cm and ≥11.5 cm), and severe (<11.5 cm); and for ages 60–120 months: normal (≥14.5 cm), moderate (<14.5 and ≥13.5 cm), and severe (<13.5 cm). ^3^ Classification of anemia was based on age-specific hemoglobin cut-offs: for 6–23 months: mild (9.5–10.4 g/dL), moderate (7.0–9.49 g/dL), and severe (<7.0 g/dL); for 24–59 months: mild (10.0–10.9 g/dL), moderate (7.0–9.9 g/dL), and severe (<7.0 g/dL); and for 5–11 years: mild (11.0–11.4 g/dL), moderate (8.0–10.9 g/dL), and severe (<8.0 g/dL). ^4^ Feeding difficulties reported by caregiver. ^5^ Risk for feeding difficulties as determined by the study feeding screening tool. ^6^ Hemoglobin was not measured in children with reported infections.

**Table 3 nutrients-18-00200-t003:** Association of risk for feeding difficulties with undernutrition in children with disabilities enrolled in the study ^1^.

	Model 1 (Unadjusted)		Model 2 (Demographics) ^2^		Model 3 (Demographics + Health) ^3^	
	OR (95% CI)	*p*-Value	AOR (95% CI)	*p*-Value	AOR (95% CI)	*p*-Value
Underweight (WAZ) Ages 0–120 months *N* = 402	1.84 (1.20, 2.83)	**0.005**	2.34 (1.27, 4.31)	**0.007**	2.28 (1.23, 4.24)	**0.009**
Stunting (L/HAZ) Ages 0–120 months *N* = 337	1.79 (1.12, 2.86)	**0.015**	2.39 (1.24, 4.59)	**0.009**	2.46 (1.26, 4.79)	**0.008**
Wasting (WL/HZ) Ages 0–59 months *N* = 319	2.79 (1.56, 5.00)	**0.001**	2.47 (1.12, 5.43)	**0.025**	2.43 (1.10, 5.35)	**0.028**

AOR: adjusted odds ratio; CI: confidence interval; L/HAZ: length/Height-for-age *z*-score; OR: odds ratio; WAZ: weight-for-age *z*-score; WL/HZ: weight-for-length/height *z*-score. ^1^ Binary logistic regression was used to identify the association of risk for feeding difficulties (no/yes) with undernutrition indicators. *p*-values shown in bold are statistically significant (<0.05). ^2^ Model 2 adjusted for sex, age, and type of health condition (cleft lip and/or palate/cerebral palsy/other developmental disabilities). ^3^ Model 3 adjusted for sex, age, type of health condition, presence of infection (no/yes), and number of conditions (one/two or more).

**Table 4 nutrients-18-00200-t004:** Qualitative themes and illustrative quotations from families of children with disabilities enrolled in the study and key stakeholders.

Themes	Sub-Themes	Illustrative Quotations
**Barriers**	Lack of access to food	“Sometimes it becomes difficult to access the variety of food. Like some of them, they need money to buy. And other times, we fail to access the source.” Father of one-year old boy with cerebral palsy
	Poor access to services	“Because of the distance, I come from very far from this hospital and so if I am to come here, I will be needing a lot of money in terms of transport to reach here which makes it hard.” Mother of four-month-old girl with cleft lip and palate “Children with disabilities may struggle to access adequate healthcare, food, education, and support services” Stakeholder, Residential Care Facility
	Feeding difficulties	“My child has not yet been operated on so some food/milk comes out from the nose.” Mother of six-month old boy with cleft lip and palate
	Stress	“I found a hard time with my child because I had never seen any birth defect like that before; I was very devastated and shocked.” Mother of nine-month-old boy with cleft lip and palate
**Enabling factors**	Access to food	“I make sure I buy enough and variety depending on the money that I have.” Father of one-month old boy with cleft lip and palate
	Access to services	“I always have the transport money to come for reviews when given.” Mother of one-month-old boy with cleft lip and palate “One strength that I see is government support of providing free medical services at public hospitals where every child is able to access such services through social workers, Village health teams and other agencies” Stakeholder, NGO
	Nutrition knowledge	“I have learnt to feed my child, prepare his feeds and to look after my child well.” Mother of one-month-old boy with cleft lip and palate
	Feeding skills	“My child has a good appetite and feeds well. Secondly, we were taught here on how to prepare meals for such as our child and their sitting positions.” Mother of six-month-old boy with cleft lip and palate
	Economic stability	“My husband provides for us and buys variety of foods for the baby to give.” Mother of six-month-old boy with cleft lip and palate
**Recommendations**	Improving access to services	“I will advise them to try and support these children by providing services and basic needs at a free cost to all children with disabilities.” Mother of six-month-old boy with cleft lip and palate
	Raising awareness	“I will advise them to carry out sensitizations and community awareness among people on radios and televisions to make people aware of such disabilities among children and how to help them.” Mother of eight-month-old girl with cleft lip and palate “Conduct more research and use such data for advocacy and resource mobilization, funds which will be used to support these children.” Stakeholder, Government
	Support for families	“I would advise the policy makers to prioritize counselling services and emotional support to parents with children with disabilities because this experience can be overwhelming for them.” Mother of one-year-old boy with cleft lip and palate “I would advise the government to try and set up income generating projects like poultry, rearing for caregivers so as for them to make a living.” Stakeholder, Residential Care Facility

## Data Availability

The datasets presented in this article are not readily available because the data are part of ongoing analysis. Requests to access the datasets should be directed to Zeina Makhoul at zeina@spoonfoundation.org.

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
