# Peer review of "Undernutrition and Feeding Difficulties Among Children with Disabilities in Uganda: A Cross-Sectional Study"

_nutrients, 2026, doi:10.3390/nu18020200_

Round 1

Reviewer 1 Report

Comments and Suggestions for Authors

This manuscript addresses the relation between feeding difficulties and undernutrition among children in disabilities in Uganda. Children with disabilities are clearly at risk for undernutrition. Strengths of the manuscript include a comprehensive background on current findings among children with disabilities in low-and-middle income countries, clear objectives related to the prevalence of undernutrition among children with disabilities enrolled at CoRSU, perceived barriers to adequate nutrition, and gaps and opportunities in nutrition services and policies for children with disabilities.  The manuscript is well-written and well organized, with links to global policies for children with disabilities. There are several issues that the authors could consider:

  1. In Uganda, the prevalence of cerebral palsy is probably higher than the prevalence of cleft lip and palate, whereas in this sample, the prevalence of cleft lip and palate is higher. Please provide a few more details about the Comprehensive Rehabilitation Services for People with Disability in Uganda Rehabilitation Hospital (CoRSU), including the eligibility criteria for children, whether children are referred, whether CoRSU specializes in specific services in addition to surgical repair for cleft lip and palate, and how services are funded. This information would be very helpful in evaluating the generalizability of the sample and findings.
  2. It would be helpful to provide more information on the feeding practices tool and perhaps to include it in the supplementary material. The items for children under 12 months include holding the infant’s head during feeding and burping. These are age-specific activities that are no longer appropriate once the infant has head control and can sit. Thus, parents of infants nearing 12 months may not endorse them because the infant no longer requires head holding or burping. Please clarify how responsive feeding was measured and if the measure is robust enough to be included in the abstract (e.g., responsive care). Please report if the tool has been validated and if not, please add it as a limitation.
  3. The methods of measuring feeding difficulties and risk of feeding difficulties are very briefly described in the Methods. It would be helpful to have a clear description of how the two areas were operationalized. It would also be helpful for others seeking to measure feeding difficulties and risk of feeding difficulties.
  4. Infants with cleft lips or palates often have difficulty sucking, as the authors report. In some situations, mothers pump breastmilk and deliver it through a modified or adapted bottle that requires less sucking than at the breast. The paper could be clarified by referring to the milk the infants receive as either formula or human milk and the mode of feeding as either breast or bottle or something else. By referring to breastfeeding, it is not clear how the milk is delivered.
  5. Please clarify the objectives for the qualitative study. The paper refers to the methods, but not the type of questions that were asked. Qualitative analyses are often very helpful in understanding the drivers behind the quantitative data. Here the focus on barriers, enabling factors, and recommendations is very useful information. The attempt to analyze the quantitative data from the 32 respondents is less useful – would consider removing.  
  6. It appears that data in the home environment, including socioeconomic status, food security, or maternal education are not available. These characteristics are often related to children’s nutritional status. The lack of these data is a limitation and could recommended for inclusion in future studies.
  7. This paper clearly adds to the existing information on undernutrition and feeding difficulties among children with disabilities. There is no need to include “this is the first…” Science is an iterative process – investigators build on findings from others, as these authors have done, and moving forward, others will build on these findings.
  8. It would be helpful to clarify the recommendation for national nutrition programs and policies. Children with disabilities are clearly at risk of undernutrition and many may require special services. At the same time, national nutrition programs are often under- funded, under-implemented, and lack the expertise needed to address the needs of children with disabilities. Please note if there are existing models or suggest innovative collaborative models that could be evaluated in pilot programs.

Author Response

Dear Reviewer,

Thank you very much for your thoughtful and constructive feedback. We have carefully considered your comments and believe we have addressed them comprehensively. Our responses are provided in red font following each of your comments. Please find the corresponding revisions/corrections in track changes in the re-submitted files. We truly appreciate the time and effort you dedicated to reviewing our work.

Wishing you Happy Holidays and a wonderful start to the New Year!

This manuscript addresses the relation between feeding difficulties and undernutrition among children in disabilities in Uganda. Children with disabilities are clearly at risk for undernutrition. Strengths of the manuscript include a comprehensive background on current findings among children with disabilities in low-and-middle income countries, clear objectives related to the prevalence of undernutrition among children with disabilities enrolled at CoRSU, perceived barriers to adequate nutrition, and gaps and opportunities in nutrition services and policies for children with disabilities.  The manuscript is well-written and well organized, with links to global policies for children with disabilities.

Thank you! Your positive feedback is much appreciated.

There are several issues that the authors could consider:

Comment 1: In Uganda, the prevalence of cerebral palsy is probably higher than the prevalence of cleft lip and palate, whereas in this sample, the prevalence of cleft lip and palate is higher. Please provide a few more details about the Comprehensive Rehabilitation Services for People with Disability in Uganda Rehabilitation Hospital (CoRSU), including the eligibility criteria for children, whether children are referred, whether CoRSU specializes in specific services in addition to surgical repair for cleft lip and palate, and how services are funded. This information would be very helpful in evaluating the generalizability of the sample and findings.

Response 1: Thank you for pointing this out. We agree that, at the population level in Uganda, cerebral palsy (measured as existing cases among children 0-17 years old) is more prevalent than cleft lip and/or palate (typically measured at birth through hospital-based and birth registry studies). The higher proportion of children with cleft conditions in our sample reflects the service profile of CoRSU rather than underlying population prevalence. CoRSU functions as a national referral center with a strong focus on surgical and rehabilitative care for children with cleft lip and/or palate, in addition to providing broader pediatric orthopedic, rehabilitation, nutrition, and psychosocial services. As a result, children with cleft conditions are more likely to be represented in this hospital-based sample.

To support interpretation of the findings and their generalizability, we have expanded the Methods section to provide additional detail on CoRSU’s eligibility criteria, referral pathways, range of services, and funding mechanisms (Lines 115-129). We hope this added context clarifies the composition of the study sample and situates the findings appropriately.

Comment 2: It would be helpful to provide more information on the feeding practices tool and perhaps to include it in the supplementary material. The items for children under 12 months include holding the infant’s head during feeding and burping. These are age-specific activities that are no longer appropriate once the infant has head control and can sit. Thus, parents of infants nearing 12 months may not endorse them because the infant no longer requires head holding or burping. Please clarify how responsive feeding was measured and if the measure is robust enough to be included in the abstract (e.g., responsive care). Please report if the tool has been validated and if not, please add it as a limitation.

Response 2: Thank you for this feedback. We have added additional detail in the Methods section (lines 230-239) describing the feeding practices tool and have included the full set of questions as supplementary material (line 231).

We also appreciate the points raised regarding burping infants and providing head support during feeding. We agree that these practices are typically needed until approximately 4-6 months of age, after which many infants begin to burp independently as their digestive systems mature (sometimes earlier than 4 months) and are able to sit with good head control and minimal support. However, this developmental trajectory may differ for infants with developmental delays and disabilities. In response, we revised the Results (lines 425-430) and Figure 1 to report these two practices specifically for infants under 6 months of age. We selected 6 months as the upper age limit given that our sample includes infants with cleft lip and/or palate – who may require burping for a longer period – as well as infants with disabilities who may have motor delays. Results were similar regardless of whether the age range was defined as 0-6 months or 0-12 months, as a large proportion of infants under 12 months in the sample were younger than 6 months.

Regarding responsive feeding, while our abstract originally stated that “most caregivers reported using responsive feeding practices”, we did not create a single indicator of responsive feeding and do not report a composite measure of responsive feeding. Instead, we report on each individual practice separately (e.g., supporting the head, responding to feeding cues, interactive engagement, pacing, use of age-appropriate tools). These practices collectively reflect responsive feeding, but the measure should be interpreted as descriptive rather than a formal composite indicator. To avoid misinterpretation, we revised the abstract (line 32), clarified this approach in the Methods section (lines 237-239), and added a WHO/UNICEF reference on responsive feeding.

Finally, we note that the structured questionnaire used in this study has not been formally validated. This has now been acknowledged as a study limitation (lines 771-772).

Comment 3: The methods of measuring feeding difficulties and risk of feeding difficulties are very briefly described in the Methods. It would be helpful to have a clear description of how the two areas were operationalized. It would also be helpful for others seeking to measure feeding difficulties and risk of feeding difficulties.

Response 3: We added additional information in the Methods section (lines 215-223) describing how we screened for risk of feeding difficulties and have included the full set of screening questions as supplementary material (line 209). We also clarify that the tool is a screening instrument and therefore measures risk of feeding difficulties, rather than diagnosing feeding difficulties themselves (225-226).

Comment 4: Infants with cleft lips or palates often have difficulty sucking, as the authors report. In some situations, mothers pump breastmilk and deliver it through a modified or adapted bottle that requires less sucking than at the breast. The paper could be clarified by referring to the milk the infants receive as either formula or human milk and the mode of feeding as either breast or bottle or something else. By referring to breastfeeding, it is not clear how the milk is delivered.

Response 4: Thank you for highlighting this important point. We added clarification in the Methods section (lines 209-212) on how breastfeeding was measured, including distinctions between direct breastfeeding and feeding expressed breast milk. We also revised the Results section (lines 483 & 494) and expanded the Discussion to more fully address direct breastfeeding, expressed breast milk, and infant formula, both in relation to general feeding practices at CoRSU (lines 670-675) and what we observed in our study sample (lines 675-680).

In addition, we updated the proportion of infants with cleft lip and/or palate under 6 months who were classified as exclusively breastfed (line 681). This update includes infants who exclusively received breast milk via bottle, spoon, or cup, in line with the UNICEF definition of exclusive breastfeeding. Although this revision increased the estimate, it remains substantially lower than national figures. Finally, we clarified throughout the manuscript whether references to breastfeeding indicate direct breastfeeding, feeding expressed breast milk, or both.

Comment 5: Please clarify the objectives for the qualitative study. The paper refers to the methods, but not the type of questions that were asked. Qualitative analyses are often very helpful in understanding the drivers behind the quantitative data. Here the focus on barriers, enabling factors, and recommendations is very useful information. The attempt to analyze the quantitative data from the 32 respondents is less useful – would consider removing.

Response 5: We reiterated the objectives specific to the qualitative component of the analysis in the Methods section (lines 253-255) and added details on the types of questions asked of caregivers (lines 258-261) and stakeholders (lines 263-265).

Thank you for the suggestion to consider removing the quantitative analysis based on the 32 respondents (lines 515-518). We carefully considered this option but chose to retain the analysis to help illustrate how contextual factors interact with health conditions in shaping feeding-related outcomes. While we acknowledge that the small sample size limits robust triangulation - an issue we note as a limitation (lines 517-518 & 754) - the quantitative findings provide a useful indication of consistency with the qualitative data and help contextualize the qualitative results.

Comment 6: It appears that data in the home environment, including socioeconomic status, food security, or maternal education are not available. These characteristics are often related to children’s nutritional status. The lack of these data is a limitation and could recommended for inclusion in future studies.

Response 6: This was highlighted as a limitation (line 779-781) and we now added a recommendation for investigating other potential determinants of undernutrition in future research (line 796-797).

Comment 7: This paper clearly adds to the existing information on undernutrition and feeding difficulties among children with disabilities. There is no need to include “this is the first…” Science is an iterative process – investigators build on findings from others, as these authors have done, and moving forward, others will build on these findings.

Response 7: We rephrased the sentence to shift from a “first” claim to a “value-added” framing (line 569-572).

Comment 8: It would be helpful to clarify the recommendation for national nutrition programs and policies. Children with disabilities are clearly at risk of undernutrition and many may require special services. At the same time, national nutrition programs are often under- funded, under-implemented, and lack the expertise needed to address the needs of children with disabilities. Please note if there are existing models or suggest innovative collaborative models that could be evaluated in pilot programs.

Response 8: These are very important points and help to elucidate key barriers. In response, we expanded our recommendation to mainstream disability within nutrition practices by adding more context-specific detail for Uganda (lines 788-792). We also incorporated relevant global guidance and frameworks on strengthening disability support (lines 589–595). We hope that these additions address the reviewer’s concerns while remaining within the scope of our study.

Reviewer 2 Report

Comments and Suggestions for Authors

The research topic undertaken by the authors addresses the important issue of undernutrition and nutritional disorders among children with disabilities, using Uganda as an example. This is a problem prevalent in many low- and middle-income countries, especially in Africa. Therefore, a closer understanding of the determinants of this phenomenon has significant cognitive and practical value. The authors enrolled 428 children under the age of 10 who represented various disabilities and health conditions. A strength of the presented study is its reliance on objective data obtained from anthropometric measurements, anemia screening, assessment of feeding practices, and analysis of qualitative data

However, a weakness of the reviewed article is its overly general conclusions. In principle, they could apply to many low- and middle-income countries. I encourage the authors to emphasize in their conclusions the specificity of the situation in Uganda in the scope studied compared to other African countries. Furthermore, they should propose specific solutions aimed at mitigating the negative effects of undernutrition and nutritional disorders among children with disabilities. I also noticed that in the 'References' section there are no actual publications on the topic, such as, for example, 1. Yan Ch. et al. Co-occurrence and associated factors of developmental delay, functional disability, and undernutrition in early childhood: evidence from 15 population-based surveys in sub-Saharan African countries. eClinicalMedicine, 2025;85: 103305, Published Online 14 June. 2025, https://doi.org/10.1016/j.eclinm.2025.103305. 2. Hume-Nixon M., Kuper H. The association between malnutrition and childhood disability in low- and middle-income countries: systematic review and meta-analysis of observational studies. Tropical Medicine and International Health, 2018, vol. 23 no. 11, pp. 1158–1175. doi:10.1111/tmi.13139. Including these publications in the text of the article would be helpful for making appropriate comparisons in the 'Discussion' section. In conclusion, the article requires some corrections and additions, which I mentioned above.

Author Response

Dear Reviewer,

Thank you very much for taking the time to review this manuscript. We have carefully considered your comments and believe we have addressed them comprehensively. Please find the detailed responses below (in red font) and the corresponding revisions/corrections in track changes in the re-submitted files.

Wishing you Happy Holidays and a wonderful start to the New Year!

The research topic undertaken by the authors addresses the important issue of undernutrition and nutritional disorders among children with disabilities, using Uganda as an example. This is a problem prevalent in many low- and middle-income countries, especially in Africa. Therefore, a closer understanding of the determinants of this phenomenon has significant cognitive and practical value. The authors enrolled 428 children under the age of 10 who represented various disabilities and health conditions. A strength of the presented study is its reliance on objective data obtained from anthropometric measurements, anemia screening, assessment of feeding practices, and analysis of qualitative data.

Thank you for your positive feedback.

However, a weakness of the reviewed article is its overly general conclusions.

Comment 1: In principle, they could apply to many low- and middle-income countries. I encourage the authors to emphasize in their conclusions the specificity of the situation in Uganda in the scope studied compared to other African countries.

Response 1: Thank you for your feedback. We took care to avoid overgeneralizing our findings by consistently using qualifiers such as “in our sample” and “among children with disabilities accessing care at CoRSU” throughout the manuscript. While our findings and recommendations – considered alongside regional and global evidence – may be relevant to other low- and middle-income countries, we agree that conclusions should remain within the scope of the study. Accordingly, we revised the language in the abstract (line 35), the opening of the Discussion (lines 5690-572), and the Conclusion (line 804) to clarify that our findings are specific to Uganda.

Comment 2: Furthermore, they should propose specific solutions aimed at mitigating the negative effects of undernutrition and nutritional disorders among children with disabilities.

Response 2: We appreciate the reviewer’s suggestion to address specific solutions to mitigate the negative effects of undernutrition among children with disabilities. While being mindful of the scope of this paper, we made several revisions to strengthen this section. Specifically, we expanded our recommendation to mainstream disability within nutrition practices by adding more Uganda-specific detail (lines 788-792), incorporated relevant global guidance and frameworks on strengthening disability support (lines 589-595), and added general recommendations to mitigate undernutrition among children with disabilities, particularly those with feeding difficulties (lines 759-763). Presenting specific interventions and implementation strategies, however, is beyond the scope of the current study.

Comment 3: I also noticed that in the 'References' section there are no actual publications on the topic, such as, for example, 1. Yan Ch. et al. Co-occurrence and associated factors of developmental delay, functional disability, and undernutrition in early childhood: evidence from 15 population-based surveys in sub-Saharan African countries. eClinicalMedicine, 2025;85: 103305, Published Online 14 June. 2025, https://doi.org/10.1016/j.eclinm.2025.103305. 2. Hume-Nixon M., Kuper H. The association between malnutrition and childhood disability in low- and middle-income countries: systematic review and meta-analysis of observational studies. Tropical Medicine and International Health, 2018, vol. 23 no. 11, pp. 1158–1175. doi:10.1111/tmi.13139. Including these publications in the text of the article would be helpful for making appropriate comparisons in the 'Discussion' section.

Response 3: Thank you for providing these references. The Hume-Nixon et al. paper was already cited in our manuscript (reference #4, lines 47, 632). However, despite our thorough literature review, we had not included the Yan et al. study, which is highly relevant. We have now reviewed it and incorporated pertinent findings into the Discussion section (lines 635-639). Notably, Uganda was not among the 15 countries included in the Yan et al. analysis either, further highlighting the need for more data from Uganda.

In conclusion, the article requires some corrections and additions, which I mentioned above.